# Bearing Fault Diagnosis Using Lightweight and Robust One-Dimensional Convolution Neural Network in the Frequency Domain

**DOI:** 10.3390/s22155793

**Published:** 2022-08-03

**Authors:** Mohammed Hakim, Abdoulhadi A. Borhana Omran, Jawaid I. Inayat-Hussain, Ali Najah Ahmed, Hamdan Abdellatef, Abdallah Abdellatif, Hassan Muwafaq Gheni

**Affiliations:** 1Department of Mechanical Engineering, College of Engineering, Universiti Tenaga Nasional, Jalan IKRAM-UNITEN, Kajang 43000, Selangor, Malaysia; 2Department of Mechanical and Mechatronic Engineering, Faculty of Engineering, Sohar University, Sohar P.C 311, Oman; aomran@su.edu.om; 3College of Graduate Studies, Universiti Tenaga Nasional, Jalan IKRAM-UNITEN, Kajang 43000, Selangor, Malaysia; jawaid@uniten.edu.my; 4Department of Civil Engineering, College of Engineering, Universiti Tenaga Nasional, Jalan IKRAM-UNITEN, Kajang 43000, Selangor, Malaysia; mahfoodh@uniten.edu.my; 5School of Engineering-Electrical & Computer Engineering Department, Lebanese American University, Byblos 1102, Lebanon; hamdan.abdellatef@lau.edu.lb; 6Expert System and Optimization Laboratory, Department of Electrical Engineering, Faculty of Engineering, University of Malaya, Kuala Lumpur 50603, Selangor, Malaysia; abdallahh950@hotmail.com; 7Computer Techniques Engineering Department, Al-Mustaqbal University College, Hillah 51001, Iraq; hasan.muwafaq@mustaqbal-college.edu.iq

**Keywords:** deep learning, one-dimensional convolutional neural network, signal-to-noise ratio, fault diagnosis, fast Fourier transform, bearing

## Abstract

The massive environmental noise interference and insufficient effective sample degradation data of the intelligent fault diagnosis performance methods pose an extremely concerning issue. Realising the challenge of developing a facile and straightforward model that resolves these problems, this study proposed the One-Dimensional Convolutional Neural Network (1D-CNN) based on frequency-domain signal processing. The Fast Fourier Transform (FFT) analysis is initially utilised to transform the signals from the time domain to the frequency domain; the data was represented using a phasor notation, which separates magnitude and phase and then fed to the 1D-CNN. Subsequently, the model is trained with White Gaussian Noise (WGN) to improve its robustness and resilience to noise. Based on the findings, the proposed model successfully achieved 100% classification accuracy from clean signals and simultaneously achieved considerable robustness to noise and exceptional domain adaptation ability. The diagnosis accuracy retained up to 97.37%, which was higher than the accuracy of the CNN without training under noisy conditions at only 43.75%. Furthermore, the model achieved an accuracy of up to 98.1% under different working conditions, which was superior to other reported models. In addition, the proposed model outperformed the state-of-art methods as the Signal-to-Noise Ratio (SNR) was lowered to −10 dB achieving 97.37% accuracy. In short, the proposed 1D-CNN model is a promising effective rolling bearing fault diagnosis.

## 1. Introduction

The incremental expansion of the modern rotating machinery industry has necessitated the present system to perform with greater reliability and safety and, at the same time, minimal production costs and maintenance expenditures. In view of this, fault diagnosis is a crucial element that can improve product efficacy and reduce the risk of accidental hazards in sophisticated mechanical systems [1]. The rolling element bearing is a vital component in rotating machinery fault diagnosis, where bearing failures contribute roughly 45–55% of the total mechanical equipment failures [2,3,4,5], while bearing faults account for 90% of small rotary machine failures [6,7]. The early detection of motor faults and correct diagnosis performance, particularly during complex and changeable load and working conditions, is therefore essential to avoid heavy financial loss and prevent catastrophic consequences [8,9,10].

With the tremendous development in the fields of artificial intelligence, machine learning has become an essential technique and approach utilised in many areas, including fault diagnosis [11,12], computer vision [13], structural engineering [14], machine health monitoring [15], electricity production [16], etc.

Over the years, three main steps have been considered for Machine Learning (ML)-based fault detection methods of rolling bearings, which are: (1) obtaining vibrational signals of the equipment; (2) pre-processing and extracting important feature information from the acquired signals; (3) fault diagnosis [17,18]. Vibrational signals are widely applied in signal acquisition primarily due to their ability to transmit a massive amount of information and can be easily measured [19]. Meanwhile, various rotating machinery data-driven signal processing methods, such as Fast Fourier Transform (FFT) [20], Wavelet Transform (WT) [21], Empirical Mode Decomposition (EMD) [22], and Ensemble Empirical Mode Decomposition (EEMD) [23], are utilised to evaluate the acquired signals and extract useful fault properties through the time-domain, frequency-domain, or time-frequency-domain methodologies [24]. Eventually, the retrieved signal features are applied to train machines to express the extracted features. Previously, a novel hybrid approach that integrated the Variational Mode Decomposition and Support Vector Machines (SVMs) was proposed to identify the type of rolling bearing faults [25]. In another study, an intelligent defect diagnostic approach was suggested based on the combined Singular Value Decomposition (SVD) and Local Mean Decomposition (LMD), which analysed non-linear and non-stationary vibration data, as well as the Extreme Learning Machine (ELM) [26]. Furthermore, an optimised k-nearest neighbour model was proposed according to the kernel principal component analysis and Particle Swarm Optimisation (PSO) [27], apart from other ML approaches, as described in [10,28,29].

While conventional intelligent methods functioned properly and produced accurate diagnosis outcomes, they still have two drawbacks: (1) they exhibit poor fault classification accuracy features since the signals are normally extracted through manual approaches based on prior information and diagnostic techniques, which are developed for certain fault types only and are unsuitable for other faults [30]; (2) they suffer a significant drop in model accuracy given that signals acquired in real-world industries are often subjected to environmental noise [31]. Consequently, it is imperative to construct new intelligent fault diagnosis methods that can automate fault diagnosis processes.

Besides extracting representative features from the acquired signals, Deep Learning (DL) methods are valuable for evaluating complicated and non-stationary signals, which offers the potential to address the constraints of conventional intelligent diagnosis methods [32,33]. Numerous DL models have been developed and used in rolling element-bearing fault diagnostics applications, such as Autoencoder (AE) [34], Recurrent Neural Network (RNN) [35], Deep Belief Network (DBN) [36], and Generative Adversarial Network (GAN) [37]. Among the popular DL methods is the Convolution Neural Network (CNN) [3], which assists in minimising the computational load on the network while reducing the risk of overfitting. These properties relatively enhance the accuracy and efficiency of the network, which is highlighted through its capacity to automatically extract features from images or signals [38]. Recently, Shaobo et al. [39] proposed an ensemble convolutional neural network model with improved D-S evidence fusion to study benchmark-bearing fault diagnosis. The structure of the CNN model is composed of three convolutional layers plus a full connection layer, the model was trained with two sensor signals, whose outputs are fused using the improved D-S fusion algorithm, and it showed good adaptability on the bearing fault datasets under different load conditions with multi-sensors signals. On the other hand, wanlu et al. [40] solved the problem of an imbalanced sample of the original hydraulic pump vibration signal, and integrate a one-dimensional convolutional neural network (1D-CNN) with generative adversarial networks to produce small sample size fault samples. The model showed a high-quality solution to the imbalanced sample issue during model training. Moreover, the Multi-Scale Deep CNN (MS-DCNN) was utilised to decrease the training period and network variables needed by the CNN algorithm. The model, which contained nine network layers, used a 1*1 convolution to reduce the dimensionality as well as improve the network depth and width, and greatly reduced the network parameters and the training time [41].

Based on the above-mentioned literature, DL approaches demonstrate superior performance over conventional ML techniques and offer a comprehensive alternative to fault classification. Nevertheless, the major downsides of DL approaches are as follows:Most DL and ML models perform poorly when subjected to noisy environments, where the decrease in model accuracy corresponds with the growing noise levels.Although the accuracy of the models can be increased, the structure of the models also becomes more intricate, affecting the interpretability of the real-world implementation of the models.

Thus, the proposed 1D-CNN model in this study was aimed to address the above challenges with potential contributions, which are summarised as follows:Unlike previous studies that applied only magnitude as input and discarded the phase that includes important information about the signal, this study utilised the magnitude and phase components as two separate inputs in the proposed 1D-CNN, which was trained and operated in the frequency domain. The frequency-domain representation allows a better understanding of the signal and enhances the performance in terms of accuracy and computational complexity.A lightweight four-layer 1D-CNN model was proposed with 9220 parameters, and only 2.6 M Floating-Point Operations (FLOP) were used. The model used to process the benchmarking data of Case Western Reserve University (CWRU) could achieve 100% and 99.3% accuracy with and without additive noise, respectively.The model is trained with additive noise to improve its resilience to noise. To demonstrate robustness, we show that our model, when trained with signals that have additive noise with SNR (−4~2) dB, achieves 99.3%, 98.8% and 97.3% accuracy for SNRs −6, −8 and −10, respectively.The proposed model outperforms the previous state-of-the-art works on fault-bearing detection.

Following the introduction section, Section 2 of this paper describes a brief review of the related work. In Section 3, the principles of FFT analysis and CNN are presented, while in Section 4, the proposed model is thoroughly described. Section 5 provides the analyses and evaluation of the experimental results. Finally, Section 6 presents the conclusion and recommendations of this study for future research.

## 2. Background and Related Studies

The first part of this section presents a brief literature review of past studies related to intelligent bearing fault diagnosis. The discussion highlights two fundamental components techniques that were adopted in the present study, which include the frequency-domain-based fault diagnosis method and noisy environment in rotating machinery fault diagnosis. The second part of this section provides a short introduction to the proposed method that involves FFT and several key features of the efficient 1D-CNN design and its advantages.

### 2.1. Related Studies

The problem of interference from noise signals is causing a significant issue for researchers. For instance, a study on the impact of radial internal clearance on the dynamical response of ball bearings indicates that the parameters like entropy and recurrence period density entropy refer to the complexity of the experimental signals because of the affected by noise interference [42].

In general, the time domain or the frequency domain can be used to analyse the signals. The signal properties are more distinct when time-domain signals are converted to frequency-domain signals since the latter signals are less influenced by noise compared to the former signals [43]. The presence of a fault characteristic frequency would amplify the signal component amplitudes that are associated with the characteristic component amplitudes that are associated with the characteristic fault frequency, which allows the detection of the failed bearing location through the initial vibration signal of the frequency components, which corresponds to the nature of the bearings [38,44,45]. Considering the significant noise interference at industrial sites, it is, therefore, necessary to examine the anti-noise capacity of a particular model and address the strong noise hindrance.

Numerous studies have been carried out in search of a solution to noise interference. For instance, the periodic potential to generate the adaptive stochastic resonance (PSR) is utilised in this study and aims to identify weak character signals for the diagnosis of rolling element-bearing faults. The concept of this study is that a phenomenon occurs in which the signal is enhanced, and the noise is weakened. The findings of the analysis reveal that the model is effective and is able to deal with a variety of different working scenarios. The limitation of this study is that it focuses on a specific problem for which the theoretical frequency value can be determined prior to detection. Because of this, the technique needs to be applied multiple times if the type of fault is unknown [46].

A five-layer Deep Convolutional Neural Network with a wide input kernel and small following kernels that were demonstrated to achieve high accuracy (WDCNNs) was proposed to diagnose bearing defects by utilising raw vibration signals as the input data and processing them through the WDCNN model [47]. The WDCNN performance decreased drastically when the Adaptive Batch Normalisation (AdaBN), which is a statistical algorithm that requires statistical knowledge of the entire test data, was not applied. Meanwhile, the use of Training Interference in six-layer Convolutional Neural Networks (TICNN) [48] revealed two main interferences, which include the first-layer kernel dropout with a constantly fluctuating rate and an exceptionally small training batch. Additionally, ensemble learning has been used to improve their model’s performance. Thus, the use of data augmentation facilitates the enhanced performance of the TICNN model as the working load increases.

The development of the Multi-Scale Cascade Convolutional Neural Network (MC-CNN) involved the addition of convolutional layers before the multi-scale cascade layer to generate a new signal with more recognisable information, as well as the addition of a convolutional layer with kernels of modest size and a pooling layer after the multi-scale cascade layer to restrict the abundance of neurons generation by the multi-scale signal [49]. Furthermore, the Radial Basis Function Neural Network, combined with the power spectrum of the Welch method (W-RBFNN), was developed to convert time-domain signals into power spectrum [50]. The method could also simultaneously remove the effects of the initial phase difference and minimise the noise effect.

While these approaches have achieved significant progress, the accuracy of these models is reduced when the noise level exceeds −4 dB. Zhou et al. [43] proposed an integrated framework via the Convolutional Neural Network and Frequency-Domain Feature Matching (CNN-FDFM) algorithm to assist in sustaining the excessive noise levels. Essential frequency features from the frequency-domain signals are captured by FDFM and retained at high accuracy with limited samples under high noise conditions. Additionally, the use of the dropout method with larger kernels in the first convolutional layer could simulate the noise input during CNN training and enhance the anti-noise ability of CNN. Nevertheless, the CNN-FDFM is unsuitable for applications under varying operating conditions as the accuracy of the model decreased by around 7% when the noise level exceeded −10 dB.

Other DL methods have been developed, such as deep neural network-based few-shot learning approach [51]. And the Stacked Inverted Residual Convolution Neural Network (SIRCNN), to avoid noise resistance [18]. The suggested neural network comprises one typical convolutional layer with a convolution kernel of 3 × 3, six inverted residual block structures, and depthwise separable convolution. Vibration spectra were created from the one-dimensional data and used as model input. Most of these methods are, however, unable to apprehend high-noise levels, limiting their possible real-world application. Hence, this study attempts to reduce the gap between theoretical findings and practical implementation.

### 2.2. Convolutional Neural Network (CNN)

CNN is one of the most prevalent DL algorithms that can reduce the number of parameters using spatial convolution relations. The model is derived from multiple layers that are fully connected to the neural network with various filter processes and a single classification process. In comparison to a fully connected network, the superior performance of CNN in numerous engineering applications is contributed by its shared weights, local connects, and pooling operators [52]. A conventional CNN framework comprises a convolutional layer, activation layer, pooling layer, and Fully Connected (FC) layer:
Convolutional layer: This layer, which utilises a class of learnable Gaussian kernel filters to convolve with the input data, generates the feature maps and can be expressed as:(1)$$x_{s}^{k}=f(\overset{J}{\underset{j=1}{\sum }}w_{js}^{k}*\;x_{j}^{k-1}+B_{s}^{k}),$$
where $X_{s}^{k}$ refers to the jth feature map at (k − 1)-th operation,$\;W_{js}^{k}$ is the k-th operation’s kernel weight parameter between the j-th input and s-th output, while $B_{s}^{k}$ represents the corresponding bias. Moreover, f (.) denotes a non-linear activation function. In addition, the Rectified Linear Unit (ReLU) is often used to perform the activation process due to its outstanding gradient efficiency, which can be termed as:(2)$$y_{ijk}^{l-1}=max\left\{0,\;x_{ijk}^{l-1}\right\},$$
where $y_{ijk}^{l-1}$ is defined as the coordinate (i,j) value in the kth feature map of the (l − 1)th layer.Activation layer: Following the convolution operation, the activation layer function is crucial for the network to obtain a non-linear expression of the input signal so that the representation ability is enhanced and permits the learned features to be further dividable. Recently, ReLU has been extensively applied as an activation unit to speed up the CNN convergence by forming more trainable weights in the shallow layer when the back-propagation learning approach is used to modify the variables. The ReLU formula is expressed as:(3)$$a^{l\left(i,j\right)}=f\left(b^{l\left(i,j\right)}\right)=\max\left\{0,b^{l\left(i,j\right)}\right\}$$
where $a^{l\left(i,j\right)}$ refers to the activation value of the output $b^{l\left(i,j\right)}$ of the convolution layer.Pooling layer: The objective of the pooling layer is to preserve spatial invariance and minimise the middle function map dimensions via the computational statistics method. The service area is first assigned by sliding a personalised pooling operation window onto the input function diagram, followed by the use of a numerical statistical approach to represent these values and minimise the resolution of the selected area. It is also crucial to select the stride parameter of the pooling layer, given its substantial impact on reducing the resolution and numerical information preservation. The maximum pooling (the maximum value in the local acceptance domain) and average pooling (average of all values in the local acceptance domain) are the frequently used pooling methods, which are expressed as follows:(4)$$y_{ijk}^{l-1}=max\left\{0,\;x_{ijk}^{l-1}\right\}$$
where 0$<i^{\prime }\leq n;0<j^{\prime }\leq m;i^{\prime }and\;j^{\prime }\in \;Z^{+}\;;n$ denotes the length of the pooling window, *m* refers to the width, while $X_{i^{\prime },j^{\prime }}^{l-1}$ denotes the covered data pooling window.
(5)$$y_{i}^{l}+{\;}^{1}\;\left(j\right)=\underset{\left(j-1\right)W+1\leq t\leq jW}{\mathrm{average}}\cdot \;\{U_{i}^{l}\left(t\right)\}$$
where W represents the pool area width; $U_{i}^{l}\left(t\right)$ denotes the tth neuron value in the ith eigenvector of the lth layer, and the $t\;\in \left[\left(j-1\right)w+1,jw\right];\;y_{i}^{l}+$ ^1^ (j) corresponds to the l+1 neuron value.FC layer: The final layer is designed to complement its non-linear input. The completely connected layer fitting operation is expressed as follows:(6)$$Y=f_{F}\left(WX_{i}^{l}+b\right)$$
where Y refers to the output, W represents the full connection matrix, and $X_{i}^{l}$ defines the output of the upper layer. Additionally, f_F_ denotes the activate function, while the number of formed categories is nearly equal to the output channel of the final FC layer. Figure 1 illustrates the flow process of CNN.

### 2.3. Fast Fourier Transform (FFT)

FFT is a useful analytical tool to remodel periodical waveforms via series harmonics in which the harmonic frequency is denoted as a multiple of fundamental. The FFT and its inverse formula are expressed as follows:(7)$$F\left(\omega \right)=\int_{-\infty }^{\infty }f\left(t\right)e^{i\omega t}dt$$
(8)$$f\left(t\right)=\frac{1}{2\pi }\int_{-\infty }^{\infty }F\left(\omega \right)e^{i\omega t}d\omega$$
where *f*(*t*) refers to a given time-domain signal and *F*(*ω*) represents the FFT of the f(t) in the frequency domain. Despite that FFT is an ideal instrument to determine the natural structural frequencies, the method is unable to indicate the time information when a particular frequency component takes place. Naturally, the rolling element-bearing signals are non-linear and non-stationary [53]. As such, the signal structure contains hidden periodicities, which convey supplementary information.
(9)$$X^{\left(N\right)}\left(P\right)=\overset{N-1}{\underset{n=0}{\sum }}x_{n}e^{\frac{-i2\pi pn}{N}}$$
(10)$$=\overset{\frac{N}{2}-1}{\underset{g}{\sum }}x_{\left(2g\right)}e^{\frac{-i2\pi pg}{\frac{N}{2}}}+e^{\frac{-i2\pi p}{N}}\;\overset{\frac{N}{2}-1}{\underset{h}{\sum }}x_{\left(2h+1\right)}e^{\frac{-i2\pi kh}{\frac{N}{2}}}$$
(11)$$=X_{0}^{\left(\frac{N}{2}\right)}\left(P\right)+e^{\frac{-i2\pi p}{N}}\;X_{1}^{\left(\frac{N}{2}\right)}\left(P\right)$$
where *p* = 0, 1, 2, …, N − 1, and g, h = 0, 1, 2, …, $\frac{N}{2}$ − 1. Furthermore, $X_{0}^{\left(\frac{N}{2}\right)}$(*P*) represents the $\frac{N}{2}$ point of the Discrete Fourier Transform (DFT) of X(N), which is regarded as even-numbered, while $X_{1}^{\left(\frac{N}{2}\right)}\left(P\right)$ denotes the $\frac{N}{2}$ point of DFT of X(N) and is regarded as odd-numbered. Both functions are periodic and discrete. Apart from that, consider the following:(12)$$W_{N}=e^{\frac{-i2\pi }{N}}$$
(13)$$\mathrm{Then},\;W_{N}^{P+\frac{N}{2}}=-W_{N}^{P}$$

Here, $W_{N}^{P}$ for *p* = 0, 1, 2, …, *N* − 1 refers to the *N*th root of unity. Equations (12) and (13) are combined to derive the following:(14)$$X^{\left(N\right)}\left(P+\frac{N}{2}\right)=X_{0}^{\left(\frac{N}{2}\right)}\left(P+\frac{N}{2}\right)-W_{N}^{P}X_{1}\left(P\right)$$
where *p* = 0, 1, 2, …, $\frac{N}{2}$ − 1. The frequency-domain information is also derived from the signal with$\;\frac{N}{2}$ multiplications as opposed to the N complex multiplication. Consequently, the computational complexity becomes O(N log N). These vibration signals can be processed through an FFT can process by maintaining the original amplitude and phase information, splitting them into individual sinusoidal oscillations at specific frequencies [54].

## 3. Methodology

In conventional CNN methods, the network inputs are composed of non-pre-processed raw signals, which would result in low prediction accuracy irrespective of the change in the hyper-parameters. This is due to the insufficient information in the input that is useful to achieve precise classification. Therefore, it is essential to improve the prediction performance of the model by implementing information improvement techniques into the input dataset. Taking into account the extreme and rough conditions of the industrial environment with a myriad of interference, the acquired data by the sensor are severely riddled with noise.

Thus, the fault diagnosis model framework in this study was developed via a three-step approach, as shown in Figure 2. The primary step involved the vibration signal acquisition from the equipment. The second step consists of the pre-processing approach to convert the raw signals into training samples that are made up of a set of input data with associated class labels. In addition, the pre-processing step comprises two main parts, namely noise injection and FFT representation, which are further discussed in Section 3.1 and Section 3.2. Finally, a four-layer lightweight 1D-CNN for fault classification was developed.

### 3.1. Robustness Improvement with Noise Injection

Noise injection was performed on the training data to generate noisy samples with varying Signal-to-Noise Ratios (SNR) in order to improve the robustness and performance of the model to noise on the hidden noisy data. The additive Gaussian noise was added to form composite signals with varying SNR, as follows:(15)$$SNR_{dB}=10\log_{10}\left(\frac{P_{signal}}{P_{noise}}\right)$$
where $p_{signal}$ and $p_{noise}$ represent the signal power and noise power, respectively. The additive White Gaussian Noise (WGN) was employed to amplify the original signal fault, as shown in Figure 3. This was in line with past studies that stated the use of White Gaussian noise to train the network as an effective method to carry out model regularisation. Thus, enhancing the model’s robustness against input variation [55]. Moreover, the SNR of the composite noisy signal was set to 0 dB, indicating the equal value of the noise power to the original signal power, as depicted in Figure 4. The proposed CNN model was then evaluated using the noisy signals with an amplitude range of −10 to 10 dB.

### 3.2. Frequency-Domain

The signal representation using the frequency-domain exhibit several favourable characteristics over its time-domain counterpart, especially its efficient ability to identify the necessary specific frequency components [56]. Hence, FFT is used to convert time-domain vibration signals into discrete frequency components, while both FFT and DFT algorithms are applied to examine the raw frequency-domain vibration signals.

Although vibration signals are based on time-domain signals, which is the case with most DL methods, they are composed of multiple basis signals with varying phases, frequencies, and amplitudes that are frequency-domain representations of the vibration signals. Figure 5 illustrates that each fault type produces distinct vibration signals with different FFT spectrums. Moreover, signals of varying fault types possess differing dominant frequency bands. In other words, valuable information is confined to various frequency bands. Hence, FFT was employed as a simple signal pre-processing step, where the frequency representation allows an in-depth understanding of the signal, as opposed to the complex interpretation of the time domain.

Given that the FFT domain is complex-valued, the data was represented using a phaser notation, which separates the magnitude and phase into two groups of inputs for the 1D-CNN to exclude complex computations, as depicted in Figure 6. Both components provide essential features to classify the bearing fault type, where the phase ensures similar signals’ correctness and diversity, while the magnitude peaks highlight interesting features. In comparison, previous work that employed FFT only considered the magnitude and overlooked the critical information in the phase for the signal representation. The present study applied FFT to convert a fixed window size of 4096 from the time-domain input signal to the desired window size and sampling size, which is explained further in Section 4.1.

### 3.3. Development of the 1D-CNN Model

As a major DL approach, CNN offers exceptional performance in detection and classification problems. The development of CNNs involves three levels, namely the convolutional layer, pooling layer, and FC layer. Valuable features from the input data are extracted using the convolutional and pooling layers, while the FC layer is predominantly accountable for the classification process. The convolutional layer convolves the input signal using a sequence of kernels in a new receptive field. Each kernel was utilised to extract features from the input signal at specific locations by sliding the kernel with a fixed stride. The weight of the kernel was shared during the convolution procedure.

In addition, a four-layer 1D convolution network (1Conv- 1Conv- 1Pool-1FC) was employed in all experiments, as presented in Table 1. The first convolution layer filter was a 16*1 sliding window with four strides and no padding, while the second convolution layer filter was an 8*1 sliding window with four strides and no padding. Next, an adaptive average pooling layer was used to aggregate each channel into a single element by calculating the mean. Finally, the FC layer was fixed with a total of 10 neurons and a SoftMax function, based on the proposed CNN model structure shown in Figure 7.

The 1D-CNN was utilised to adaptively learn the characteristics from the raw time-domain vibration signals without prior information. The CNN input consists of a fraction of a normalised bearing fault vibration temporal signal presented in frequency, as described in Section 2. The total number of parameters and FLOPs were 9220 and 2.6 M, respectively, which was considered a lightweight CNN. Table 1 also includes the number of parameters and FLOPs applied in the proposed model.

## 4. Experimental Setup

### 4.1. Dataset Preparation and Partitioning

The CWRU [57] benchmark dataset was retrieved from the drive end of the motor at a 12 k sampling rate and was employed in this study. The data contains four different subsets, with each subset representing particular working load conditions from 0 to 3 hp. In addition, each subset consists of four different fault class labels, namely the normal class and three fault classes comprising the bearing-race (BF), inner-race (IF), and outer-race (OF) at the @6:00 centred position relative to the loading zone. The three fault classes also exhibit specific fault sizes of 0.014, 0.007, and 0.021 inches, respectively. The fault sizes were created using electro-discharge machining (EDM), which generated 10 types of classes (one normal class and nine fault classes), as detailed in Table 2, Table 3 and Table 4. Moreover, sliding windows with time-series data overlaps were applied for data augmentation to amplify the number of samples. The corresponding width of the window and the shifting step were 4096 and 290, respectively. Ultimately, each working condition contains 4000 samples, and each sample was denoted as a 4096-D vector. Figure 8 portrays the overall dataset preparation and partitioning. Table 5 shows the number of training and testing datasets for each class.

### 4.2. Training Methodology and Implementation Details

The proposed CNN model was trained for 10 epochs for each repetition. The input samples were randomly shuffled at the beginning of each learning epoch to ensure efficient network learning performance Subsequently, the standard backpropagation algorithm and the Stochastic Gradient Descent (SGD) optimisation were applied with a momentum and learning rate of 0.9 and 0.01, respectively, to train the proposed CNN model throughout the experiment. Unless stated, the default settings of the open source Pytorch Framework were applied for the network training and were conducted offline. All experiments in this study were carried out using Python version 3.8 on a machine with six Core i7-9750H microprocessors, 16 GB of RAM, and NVIDIA GeForce GTX 1660 Ti with Max-Q design.

## 5. Results and Discussion

### 5.1. Performance Evaluation of Different Sampling Points

The frequency spectrum resolution may be enhanced by increasing the total number of sampling points in FFT, where the abscissa is always an integer. Normally, FFT can convert the time-domain signal with a length of N into the frequency-domain signal with a length of N/2 since it is symmetric. For instance, a sample with 4096 points would be sliced to produce a symmetric frequency spectrum with 2048 points. The shift function of the FFT was applied to shift the zero-frequency component of a Fourier transform X to the centre of the array. Figure 9a–c depicts the frequency spectrums of the inner race at 0.007 inches with varying sampling lengths.

The entire 4096 lengths of the frequency spectrum were analysed in this experiment. The sampling lengths of (a) 1024, (b) 2048, and (c) 4096 correspond to their frequency spectrums, which were composed of 1024, 2048, and 4096 points, respectively. The frequency of the k-th point is k × (fs.max/N) Hz, and these points represent various frequency features. It was observed that longer signals could produce high-resolution frequency spectrums using more points, allowing a more complete and accurate expression of the frequency-domain information. Additionally, the increase in sampling length leads to a more precise measure of the frequency features as well as more prominent discrimination between the adjacent points. Figure 10 illustrates the diagnosis results of the 1D-CNN model under varying SNRs, while Figure 11 displays the advantages of the impact of phase on fault detection at varying sampling points.

According to the findings, the accuracy of the developed 1D-CNN model was enhanced when the number of sampling points from the test samples, which were composed of original signals, was increased. When the sampling length was set at 4096, only a small shift in accuracy was recorded as the SNR of the noisy test samples decreased. In addition, the results revealed that feeding the 1D-CNN model with the phase component and magnitude component enhanced the accuracy of the fault detection, as shown in Figure 12. For instance, the use of both magnitude and phase components significantly improved the accuracy of 2048 sampling points compared to that of the 4096 sampling points for fault detection under different noise levels.

Furthermore, the conversion of time-domain signals into frequency-domain was symmetric. Following the FFT, the symmetric frequency spectrum with the length of 2048 was obtained from the sliced sample with 4096 points. Figure 13 depicts the effect of using the whole signal (4096) or symmetrical signal (2048) under different SNRs. Based on the results, the use of the entire signal achieved a greater fault detection compared to taking half of the signal (symmetric). Despite that the fault detection accuracy decreased as the noise increased, the use of the whole signal produced a more stable and less affected performance compared to the use of the half signal. The CNN would be more resilient to noise as a result of the symmetric nature of the input signal.

### 5.2. Performance Evaluation under Different Working Environments

The extremely complex working environment of mechanical systems in real-world settings can be grouped into two notable variations. First, the vibration signals are affected by noise effortlessly since noise is inevitable in industrial settings. It is also challenging and impractical to diagnose the faults in noisy environments. Second, the working load may occasionally vary relative to the required production, making it unfeasible to gather and label adequate training samples and apply a compatible robust classifier for all types of working loads. Therefore, it is crucial for feature extractors and classifiers that are trained by samples collected from one working load to adapt, learn, and classify invariant domain features. The performance evaluation of the 1D-CNN model under these two scenarios is discussed further in the following section.

### 5.3. Performance Evaluation under Noisy Environments

For the diagnosis accuracy of the proposed 1D-CNN model, the model was trained using the original data in the presence of the additive WGN to achieve a fixed SNR value (2 or −4) and randomly added to the training set. The model was also trained in the presence of additive WGN at a fixed SNR range (−4 to 2) and randomly added to the training set, as prepared through the CWRU, before being tested with the noisy data. This setting closely resembles the real-world industrial production conditions given the broad range of the noise environment, which makes it impossible to label all the training samples. The original signal of IF was added with the additive WGN, as shown in Figure 14.

The proposed 1D-CNN model was subjected to a range of noise signals from −10 dB to 10 dB. The random addition of WGN to the training set with varying SNRs (2 or −4, or 2 to −4) was performed at a fixed network structure to verify the effect of network training under different noise levels. Table 6 describes the findings of the proposed 1D-CNN model with and without the added WGN to the training set of the diagnosed noise signal.

The results indicate that the increased accuracy corresponded to the increased noise in the training set. For instance, the test accuracy was only 53.625% when the network was trained without noise addition in the training set at −10 dB. However, the test accuracy rose to 72.75% and 96.875% when the network was trained with the randomly added WGN in the training set at −10 dB under SNR of 2 and −4, respectively. Eventually, the test accuracy achieved 97.375% when the network was trained with the randomly added WGN in the training set at −10 dB under an SNR range of 2 to −4. Additionally, the proposed 1D-CNN recorded excellent performance with weak noise levels, where the model could easily obtain over 99% accuracy with an SNR value of greater than −4 dB. Interestingly, the proposed model was able to attain over 97% accuracy even when the SNR value was −10 dB after the network was trained with added WGN. Hence, training the network with WGN was an effective approach to achieving model regularisation, resulting in enhanced model robustness against varying input data.

### 5.4. Performance Evaluation under Different Load Domains

The adaptation performance of 1D-CNN under different load domains was also evaluated. Table 7 presents the scenario settings for the domain adaptation analysis. The whole signal that included both magnitude and phase components after the FFT conversion was utilised to enhance the accuracy of the proposed model. Figure 15 illustrates the performance comparison between the model using half signals (magnitude only) and the model using whole signal (magnitude and phase).

### 5.5. Performance Comparison

The rolling element bearing fault diagnosis of the proposed 1D-CNN model in this study was compared to the performance of other models from past studies, which include conventional SVM, MLP, and DNN algorithms, as well as other frequency-domain models, such as WDCNN and TICNN. Since the models were under an increased noise environment, the diagnostic accuracy declined drastically when the SNR increased, most of the previous models were ineffective when the SNR reached −4 dB. Remarkably, the accuracy of the present model was still above 97%, even at an SNR value of −10 dB. [while most of the previous models were ineffective when the SNR reached −4 dB. Remarkably, the accuracy of the present model was still above 97%, even at an SNR value of −10 dB]. Table 8 provides the performance comparison between the proposed model in this study and past models, which were under noisy conditions at an SNR of −10 dB and tested with the CWRU-bearing dataset. Figure 16 visualises the overall comparison performance of each experimental finding.

Figure 16 shows the poor performance of SVM, MLP, and DNN in terms of the domain adaptation analysis in the six scenarios with an average accuracy of 67%, 80% and 78%, respectively. In fact, the WDCNN and TICNN performed worse and never reached 90% and 95.5% accuracy, respectively. Oppositely, the proposed model in this study recorded greater precision than the other algorithms with up to 98.1%. Thus, demonstrating that the model learned from the frequency domain was more domain invariant compared to the conventional frequency features. In addition, the accuracy in each scenario improved by over 97% with high stability. As depicted in Figure 15, the proposed model exhibited a substantially greater classification accuracy when adapting from Domain A to B and B to A as well as from Domain B to C and C to B compared to from Domain A to C and C to A. Furthermore, the results showed that the model experienced greater difficulty to adapt to the new environment when the difference between the target domain and its source domain increased. The proposed model also recorded a high diagnostic accuracy of 98.8% when adapting from Domain B to C, while the accuracy was reduced by 1.5% only when adapting from Domain C to A. Besides the improved diagnosis accuracy, the proposed model achieved better stability compared to other reported models.

Referring to Figure 16 and Table 8, the WDCNN and TICNN models reported a considerable domain adaptation ability although both lacked robustness against noise interference. In contrast, the FDFM and CNN-FDFM models demonstrated robust performance on noise but were considered inappropriate for multiple working conditions. It was suggested that the robustness against noise does not easily complement the domain adaptation ability. Contrary to other models, the proposed model in this study achieved exceptional robustness to noise and concurrently excellent domain adaptation ability. Furthermore, the accuracy values in the comparative performance table demonstrated that the proposed model outperformed most of the previously published rolling element-bearing fault diagnosis models under similar and different noise environments and appears to be on par with best-performing models. Based on the overall findings, the proposed 1D-CNN model may be recommended as an effective rolling element bearing fault diagnosis.

## 6. Conclusions and Recommendations

This paper investigated the performance of an effective 1D-CNN model for bearing fault classification to address the concern over strong noise interference and diverse working conditions in industrial settings. The two-step algorithm framework was proposed, which involved the frequency-domain signal processing analysis to achieve an in-depth understanding of the signal, followed by the utilisation of two-channel input signals (magnitude and phase) in the 1D-CNN model with injected noise for the training data to generate noisy samples with high SNR under varying working conditions. The CWRU dataset was also utilised to examine the diagnosis performance of the proposed model. Based on the results, the proposed 1D-CNN model demonstrated a superior fault diagnosis capability with 100% accuracy using the normal dataset and 97.3% using the high noisy dataset. As such, the diagnosis accuracy of the proposed model was 43.75% higher compared to CNN without training under noisy conditions with SNR at −10 dB. Given that this study aimed to propose a simple model that can be easily implemented in real-time settings, this study only retrieved the public dataset with the addition of noise. Hence, future studies should consider evaluating the performance of this model using real-time data. Moreover, the extracted features from the time-domain, the frequency-domain, and the time-frequency-domain contained redundant information, which may aggravate the computation cost and lead to the curse of dimensionality. Thus, the development of a robust basic concept in signal processing is highly considered to address this issue and explain the robustness of the suggested model.

## Figures and Tables

**Figure 1 sensors-22-05793-f001:**
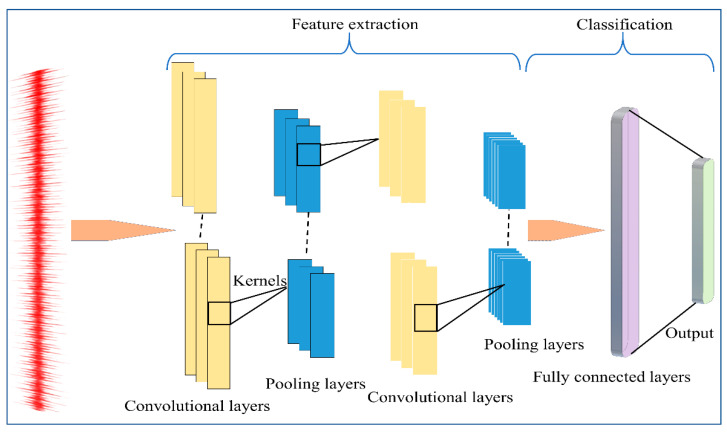
Overall design of the One-Dimensional Convolutional Neural Network (1D-CNN).

**Figure 2 sensors-22-05793-f002:**
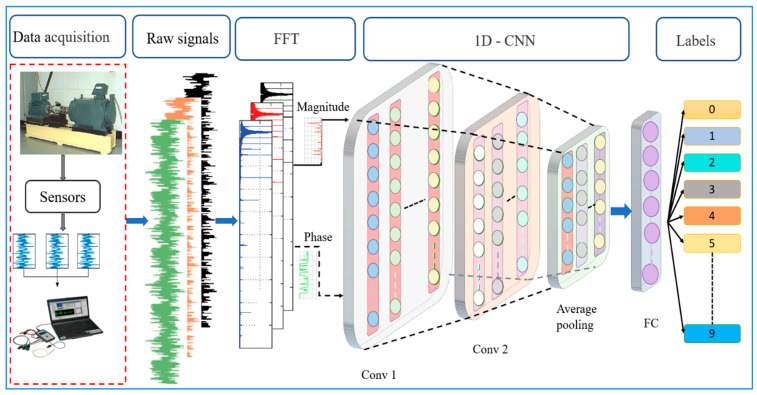
Schematic flowchart of the proposed methodology in this study.

**Figure 3 sensors-22-05793-f003:**
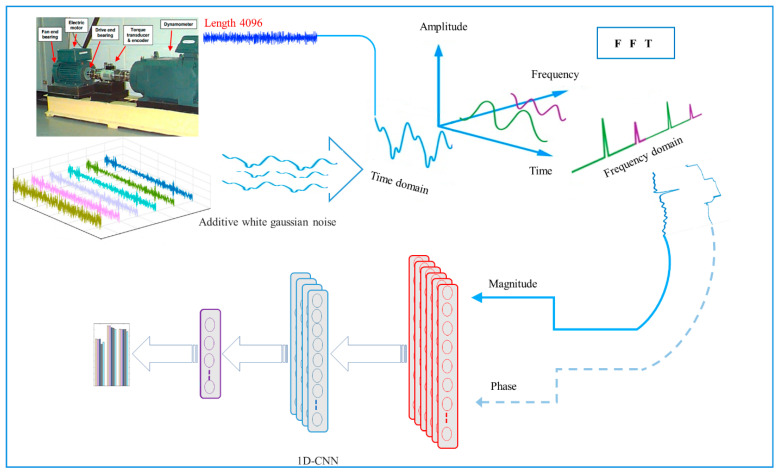
The procedure for the additive WGN injection.

**Figure 4 sensors-22-05793-f004:**
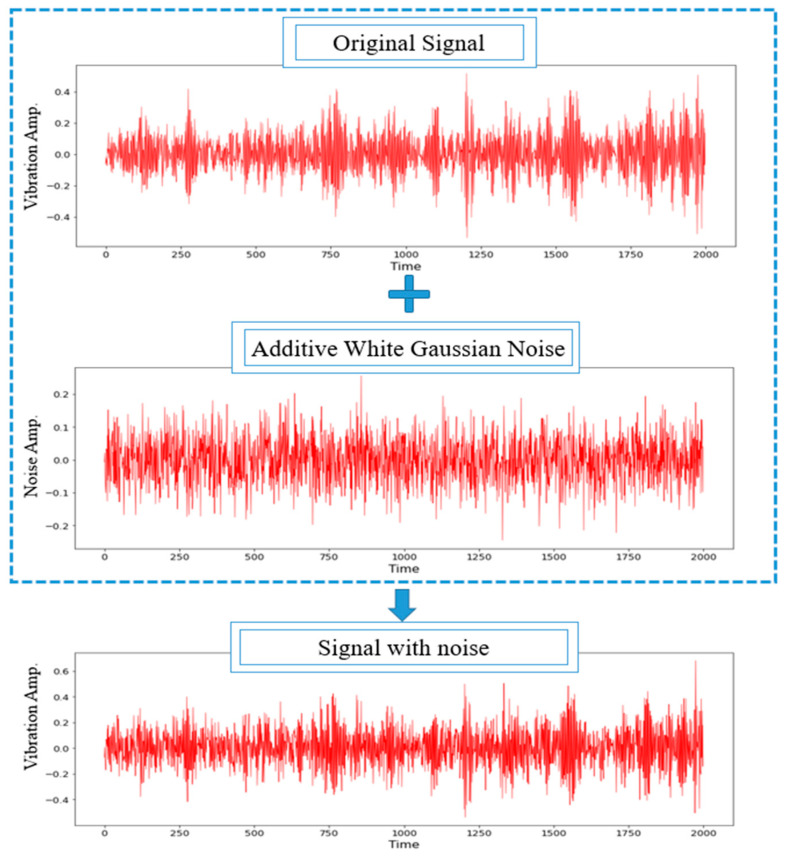
The composite noisy signal at SNR = 0 dB from the original signal of the inner race fault and the additive WGN.

**Figure 5 sensors-22-05793-f005:**
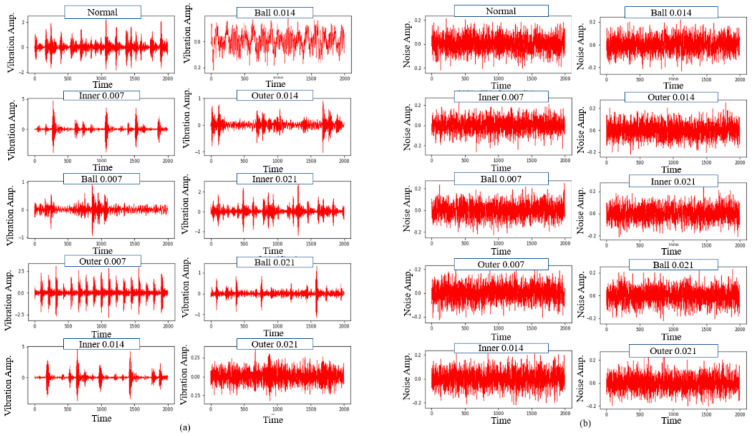
The 10 fault types of the original and noisy vibration signals based on the respective domain: (**a**) Orginal signal in time-domain, (**b**) Noise signal in time-domain, (**c**) Orginal signal in frequency-domain, (**d**) Noise signal in frequency-domain, (**e**) Combined signal (noisy signal) in time-domain, and (**f**) Combined signal (noisy signal) in frequency-domain.

**Figure 6 sensors-22-05793-f006:**
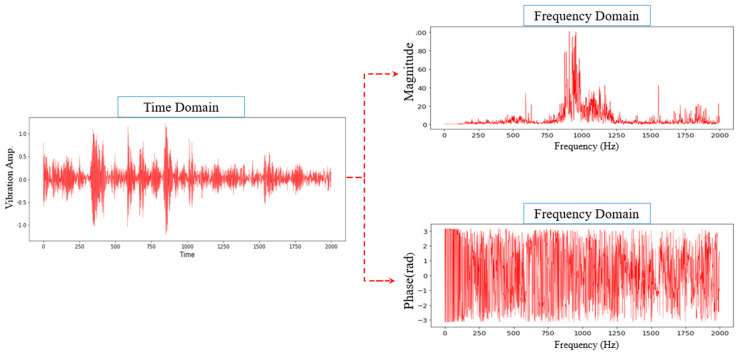
Phasor notation separating the magnitude and phase in the frequency domain.

**Figure 7 sensors-22-05793-f007:**
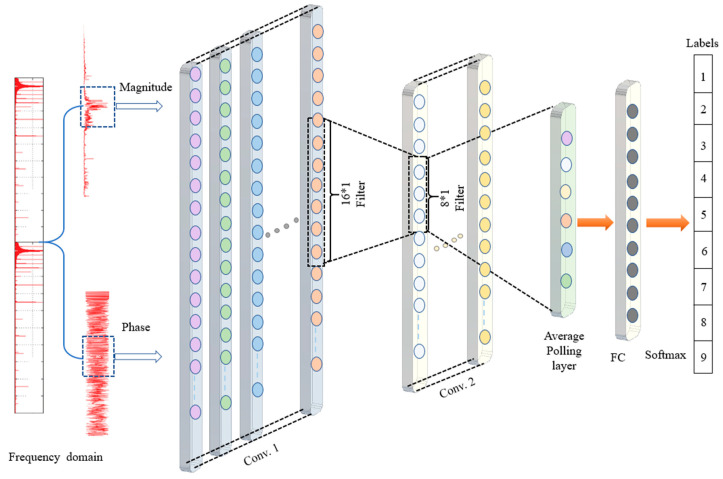
The proposed 1D-CNN structure.

**Figure 8 sensors-22-05793-f008:**
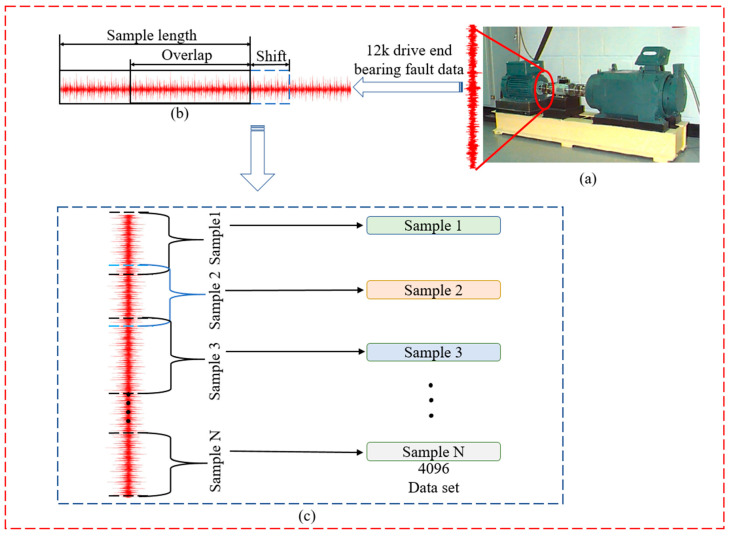
(**a**) CWRU bearing experimental platform, (**b**) Data augment with overlap, and (**c**) Vibration signal expansion mode.

**Figure 9 sensors-22-05793-f009:**
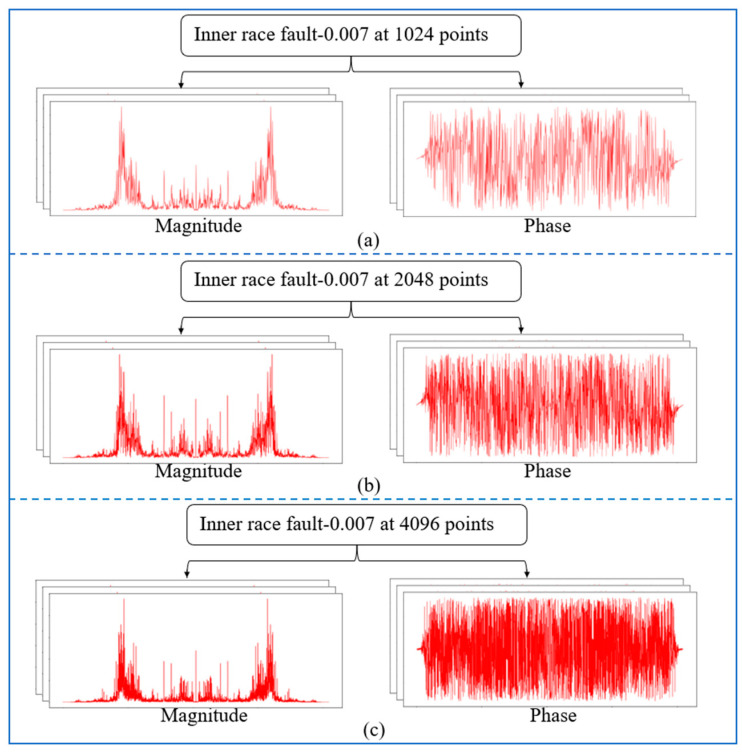
The frequency spectrums of IF at 0.007 inches with varying sampling lengths: (**a**) 1024 points, (**b**) 2048 points, and (**c**) 4096 points.

**Figure 10 sensors-22-05793-f010:**
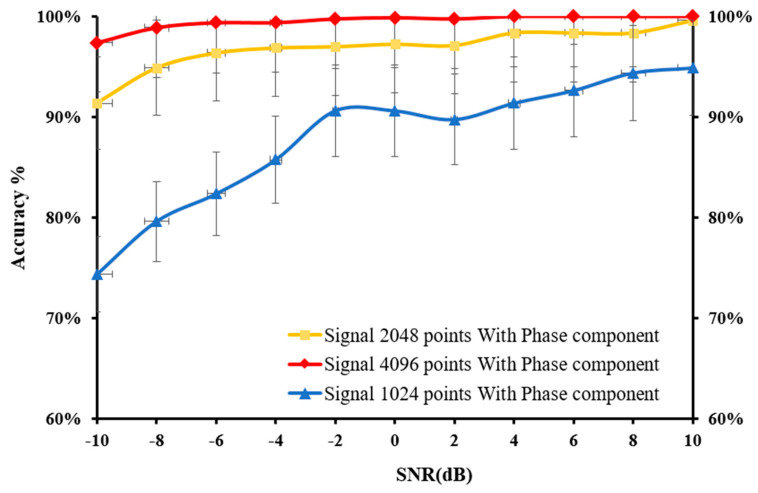
Effect of applying the phase component at various sampling points under varying SNRs.

**Figure 11 sensors-22-05793-f011:**
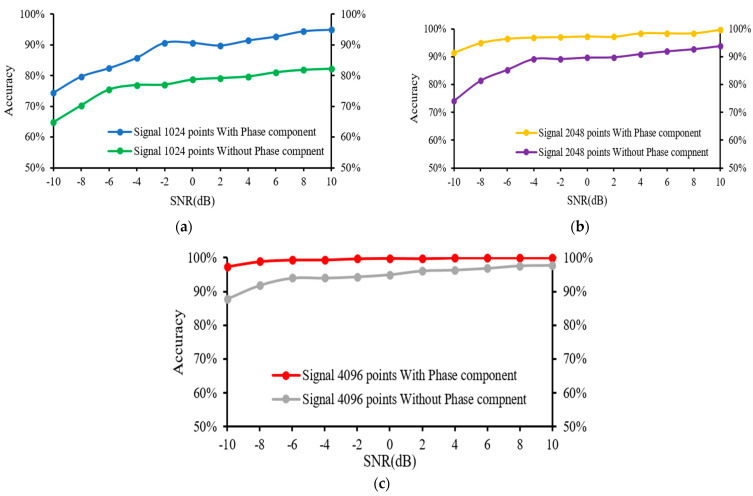
Advantage of applying the phase component on test accuracy at (**a**) 1024 points, (**b**) 2048 points, and (**c**) 4096 points.

**Figure 12 sensors-22-05793-f012:**
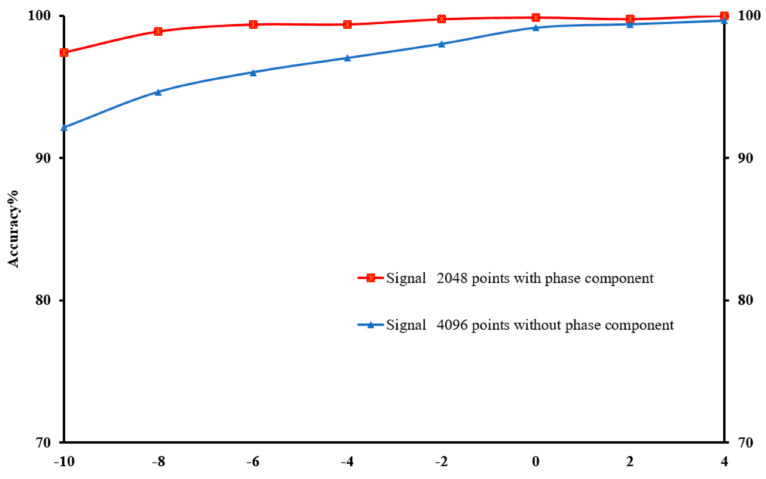
The impact of magnitude and phase components on the improved accuracy under different SNRs with a reduced number of samples.

**Figure 13 sensors-22-05793-f013:**
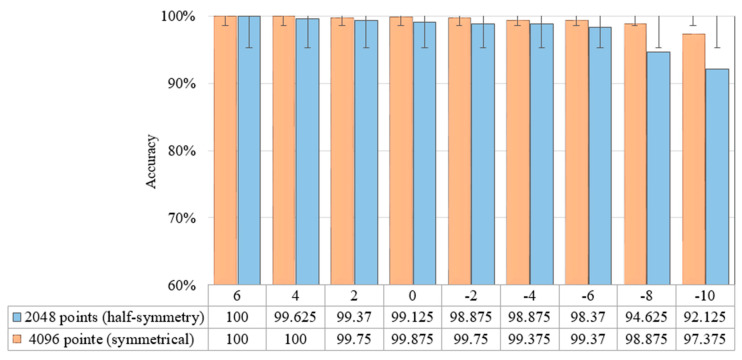
Impact of utilising whole and symmetrical signals on the fault detection under varying SNRs.

**Figure 14 sensors-22-05793-f014:**
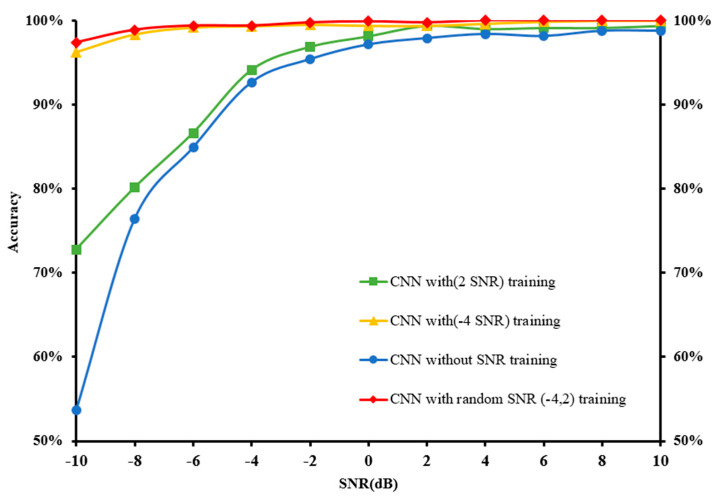
Effect of training 1D-CNN under different SNR ratios.

**Figure 15 sensors-22-05793-f015:**
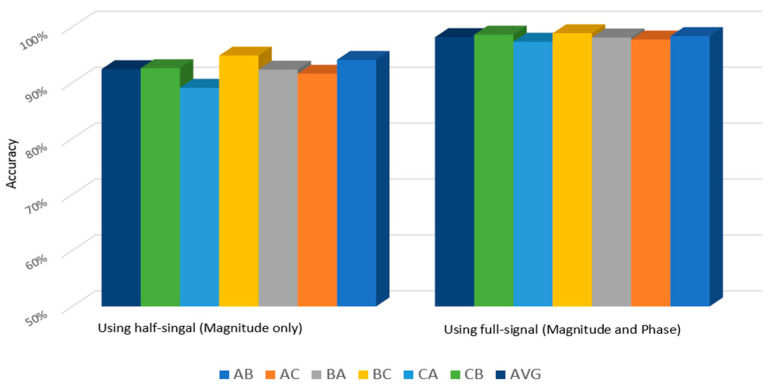
Effect of model performance using half signal and whole signal on the domain adaptation accuracy.

**Figure 16 sensors-22-05793-f016:**
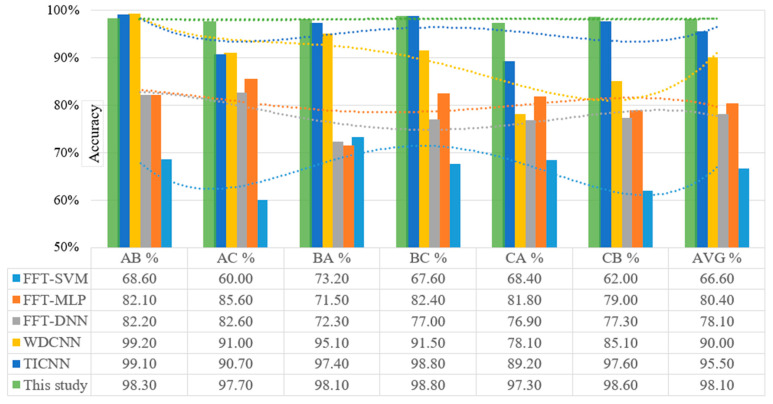
Visual comparison of the proposed 1D-CNN model of six domain shifts on Dataset A, B and C compared to recently published SVM, MLP, DNN, WDCNN, and TICNN models.

**Table 1 sensors-22-05793-t001:** The description of the proposed 1D-CNN.

Layer	Type	Kernel Size	In/Out Channels	Stride	Padding
I0	Input	-	-	-	-
C1	Conv	16	2/20	4	No
C2	Conv	8	20/50	4	No
P1	AdaptiveAvgPool1d	Adaptive	50/50	-	-
FC	Fully connected	1	50/10	-	-

**Table 2 sensors-22-05793-t002:** Dataset class description.

Motor Load (Hp)	Shaft Speed (RPM)	Normal	Bearing Fault (inch)			Inner Fault (inch)			Outer Fault (inch)		
0	1797		0.007	0.014	0.021	0.007	0.014	0.021	0.007	0.014	0.021
1	1772										
2	1750										
3	1720										

**Table 3 sensors-22-05793-t003:** The information on the rolling bearing’s structure.

No. of Rolling Elements	Ball Diameter	Outside Diameter	Inside Diameter	Thickness	Contact Angle	Pitch Diameter
9	0.3126 in.	2.0472 in.	0.9843 in.	0.5906 in.	0°	1.537 in.

**Table 4 sensors-22-05793-t004:** Characteristics of defect frequencies for bearing (6205-2RS JEM SKF).

The Frequencies Characteristic	Formula	Fault Frequencies [Hz]
Outer-race ball pass frequency (BPFO)	$\mathrm{BPFO}=\frac{nf_{r}}{2}$ $\left(1-\frac{d}{D}cos\alpha \right)$	3.5848
Inner-race ball pass frequency (BPFI)	$\mathrm{BPFI}=\frac{nf_{r}}{2}$ $\left(1+\frac{d}{D}cos\alpha \right)$	5.4152
Ball (roller) spin frequency(BSF)	$\mathrm{BSF}=\frac{Df_{r}}{2d}$ $\left(1-{\left[\frac{d}{D}cos\alpha \right]}^{2}\right)$	4.7135
Fundamental train frequency(FTF)	$\mathrm{FTF}=\frac{f_{r}}{2}$ $\left(1-\frac{d}{D}cos\alpha \right)$	0.39828

Here, *n*, *D*, *d*, $f_{r}$, α indicate the number of rolling elements, the bearing pitch diameter, the rolling element diameter, running speed in Hz and the angle of the load from the radial plane, respectively.

**Table 5 sensors-22-05793-t005:** Numbers of training and testing datasets.

Fault Location	Normal		RF			IF			OF	
Category labels	0	1	2	3	4	5	6	7	8	9
Fault diameter (inch)	0	0.007	0.014	0.021	0.007	0.014	0.021	0.007	0.014	0.021
Working condition Train	320	320	320	320	320	320	320	320	320	320
(0 HP) Test	80	80	80	80	80	80	80	80	80	80
Working condition Train	320	320	320	320	320	320	320	320	320	320
(1 HP) Test	80	80	80	80	80	80	80	80	80	80
Working condition Train	320	320	320	320	320	320	320	320	320	320
(2 HP) Test	80	80	80	80	80	80	80	80	80	80
Working condition Train	320	320	320	320	320	320	320	320	320	320
(3 HP) Test	80	80	80	80	80	80	80	80	80	80

**Table 6 sensors-22-05793-t006:** CNN training with different SNR ratios.

SNR	CNN	CNN with Fixed SNR (2)	CNN with Fixed SNR (−4)	CNN with Random SNR (−4~2)
−10	53.62	72.75	96.87	97.37
−8	76.37	80.12	98.62	98.87
−6	84.87	86.62	99.37	99.37
−4	92.62	94.12	99.62	99.37
−2	95.37	96.87	99.50	99.75
0	97.12	98.12	99.37	99.87
2	97.87	99.37	99.37	99.75
4	98.37	99.00	99.62	100
6	98.12	99.12	99.87	100
8	98.75	99.12	100	100
10	98.75	99.37	100	100

**Table 7 sensors-22-05793-t007:** Scenario settings for the domain adaptation analysis.

Domain Type	Source Domain	Target Domain	
Description	Labelled signals under one single load	Unlabelled signals under other loads	
Domain details	Training set:	Test set:	
	A	B	C
	B	C	A
	C	A	B
Target	Diagnose unlabelled vibration signals in the target domain		

**Table 8 sensors-22-05793-t008:** Comparison of accuracy between the proposed 1D-CNN model and past studies under varying SNR values.

Accuracy (%)	SNR											Ref.
	−10	−8	−6	−4	−2	0	2	4	6	8	10	
WDCNN	-	-	-	66.95	80.81	90.51	97.52	99.23	99.77	99.83	99.87	[40]
WDCNN(AdaBN)	-	-	-	92.65	97.04	98.77	99.57	99.70	99.83	99.89	99.93	[40]
TICNN	-	-	-	82.05	96.47	98.22	99.27	99.61	99.59	99.75	99.63	[39]
W-RBFNN	-	-	-	79.50	88.48	94.25	96.72	98.35	99.45	99.40	99.76	[42]
SIRCNN	-	-	96.2	99.1	99.7	100	100	100	100	100	100	[12]
FDFM	87.77	92.57	93.9	94.57	95.57	96.33	96	96.13	96.4	96.1	96.87	[36]
CNN-FDFM	93.33	96.73	99.2	99.3	99.6	99.33	99.77	99.7	9987	99.93	99.6	[36]
This study	97.37	98.87	99.37	99.37	99.75	99.87	99.75	100	100	100	100	

## Data Availability

Not applicable.

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
