# Peer review of "Bearing Fault Diagnosis Using Lightweight and Robust One-Dimensional Convolution Neural Network in the Frequency Domain"

_sensors, 2022, doi:10.3390/s22155793_

Round 1
Reviewer 1 Report
In this study, the authors proposed the One-Dimensional Convolutional Neural 29 Network (1D-CNN) based on frequency-domain signal processing. In this technique, the Fast Fourier Transform (FFT) analysis was initially utilized to convert the time-domain signals into frequency-domain signals before the signals were analyzed using two channels (magnitude and 32 phases) and then fed to the 1D-CNN. Although the results sound attractive, there are still some significant concerns that should be addressed,
1. The novelty is not sufficient. 1D-CNN has been widely used in bearing fault diagnosis. What is the main contribution of this paper?
2. In the comparison study, a. what are the features of other related methods? b. Are there noises included in the training dataset?
3. The tables and figures should be made more clear.
Reviewer 2 Report
I found your article very interesting, but in my opinion below remarks would improve your manuscript under the scientific level.
Comments and Suggestions for Authors:
1. The beginning of the Abstract is wrongly written, the Introduction is the most fitting place for the methodology description. In the Abstract please focus on your main findings “accuracy of DL” referring to the Keywords.
2. The citation style in the manuscript is incorrect.
3. Another methods, which are prone to the level of noise are recurrence analysis and stochastic resonance analysis, that is why I suggest to cite following papers discussing them by the analysis of bearing’s diagnostics:
· Liu et al. (2017), Improving the bearing fault diagnosis efficiency by the adaptive stochastic resonance in a new nonlinear system. Mechanical Systems and Signal Processing, 96, 58-76.
· Ambrożkiewicz et al. (2022), The influence of the radial internal clearance on the dynamic response of self-aligning ball bearings. Mechanical Systems and Signal Processing, 171, 108954.
4. Referring to the signal analysis, I miss the table with the bearing’s specification, i.e. its dimensions, characteristic frequencies etc. In the present form I can’t see if you detect the faults in the bearing.
5. In the “Experimental setup” you mention about fault sizes, but how do you measure it? Do you have a geometrical measurements of faults on rolling surfaces?
6. Referring to the sampling points, please specify what are the features calculated in each signal window?
Reviewer 3 Report
In this manuscript, the authors applied the FFT of the vibrational signal and obtained the frequency and phase domain information to train a 1D-CNN model for damage detection. The content is straightforward to understand. The reviewer has positive attitude about this manuscript, and only a few issues should be solved:
(1) There are some minor gramma issues in this manuscript, such as:
Line:36-41: unclear
Line121: Full stop after “operated in frequency domain” not comma
Line 128-133: unclear with gramma issue
(2) The introduction is of good quality, short, and easy to read. The downturn is that it focuses a lot on “Bearing Fault Diagnosis”. However, there are bunches of research applying similar techniques in structural health monitoring and structural damage detection. Besides, the applications of CNN in not well introduced.
https://doi.org/10.1177/1475921718757405
https://doi.org/10.1016/j.istruc.2020.12.036
https://doi.org/10.3390/s20041059
https://doi.org/10.1016/j.measurement.2022.111405
(3) The motivation for doing this research should be strengthened. The authors want to say that by introducing the phase information, the identification accuracy can be increased. Using the Hilbert-Huang Transform and wavelet transform can also achieve this goal. So, why this method?
(4) The detail of how to combine the frequency and phase domain information should be introduced in detail. For example, have data normalization techniques been used? Does frequency and phase information normalize together? Or separately?
(5) The used dataset should be introduced more clearer. A table is recommended to show the number of each original data, the augmented data, and the amount of data in the training, validation, and testing sets.
Round 2
Reviewer 2 Report
Dear Authors,
after the lecture on a revised manuscript, I see that all remarks were introduced and I'm satisfied with the answers to the given questions. I recommend the manuscript for its publishing in the present form and I wish you all the best in your future research.
Yours faithfully,
Reviewer